# Health System Response during the European Refugee Crisis: Policy and Practice Analysis in Four Italian Regions

**DOI:** 10.3390/ijerph17155458

**Published:** 2020-07-29

**Authors:** Leonardo Mammana, Chiara Milani, Paola Bordin, Lorenzo Paglione, Chiara Salvia

**Affiliations:** 1Department of Biomedical and Neuromotor Sciences, University of Bologna, 40121 Bologna, Italy; 2School of Specialization in Hygiene and Preventive Medicine, University of Florence, 50134 Florence, Italy; chiara.milani@unifi.it; 3Postgraduate School of Hygiene and Preventive Medicine, Department of Cardiac, Thoracic, Vascular Sciences and Public Health, University of Padua, 35128 Padova, Italy; paola.bordin.2@studenti.unipd.it; 4Department of Civil, Building and Environmental Engineering, Sapienza University of Rome, 00184 Rome, Italy; Lorenzo.paglione@uniroma1.it; 5Local Health Unit of Modena, 41124 Modena, Italy; C.salvia@ausl.mo.it

**Keywords:** asylum seeker, health policy, regional differences, health system, primary healthcare, vulnerability, NGOs, governance, barriers in access to care

## Abstract

The decentralization of the provision of health services at the subnational level produces variations in healthcare offered to asylum seekers (ASs) across the different Italian regions, even if they are entitled to healthcare through the national health service. The present study aims to map the healthcare path and regional policies for ASs upon arrival and identify challenges and best practices. This is a multicentric, qualitative study of migrant health policies and practices at the regional level within four Italian regions. For the analysis, a dedicated tool for the systematic comparison of policies and practices was developed. The collection and analysis of data demonstrated the presence of many items of international recommendations, even if many gaps exist and differences between regions remain. The analysis of practices permitted the identification of three models of care and access. Some aspects identified are as follows: fragmentation and barriers to access; a weakness in or lack of a governance system, with the presence of many actors involved; variability in the response between territories. The inclusion of ASs in healthcare services requires intersectoral actions, involving healthcare sectors and other actors within local social structures, in order to add value to local resources and practices, reinforce networks and contribute to social integration.

## 1. Introduction

Although the overall percentage of international migrants has only slightly increased, over the past several years, global migration has shown a growing trend in absolute numbers. Indeed, the total number of international migrants has increased from an estimated 173 million in 2000 to 272 million in 2019, primarily due to conflict, persecution, and environmental changes [1]. At the end of 2018, global displacement affected approximately 70.8 million people, more than 25 million of whom are refugees [2]. Europe has experienced an unprecedented influx of refugees, asylum seekers (ASs) and other migrants: about 1.5 million people arrived in Europe (EU) in 2015, including more than 1 million that applied for asylum, having fled countries affected by war, conflict or economic crisis [3].

Migrants are a heterogeneous group, including any person who is moving or has moved across an international border or within a state away from his or her habitual place of residence [4]. They may experience a number of health issues caused by the living conditions faced during their migratory journey and once in migrant reception centers. Moreover, the increase in the number of migrants is leading to the creation of a growing ethnically diverse population, with different languages, traditions, healthcare needs and prior levels of care. This increasing diversity will put a strain on healthcare systems, and will likely increase health inequalities [5]. Health inequalities have been defined as “differences in opportunity for different population groups which result in, for example, unequal life chances, access to health services, nutritious food, adequate housing” [6]. Although the right to health is encompassed in many European policies [7], evidence from across the EU demonstrates considerable inequalities between migrants and the local population in health and access to health services [8,9,10,11,12,13,14]. This is partly because of national legislation restricting access for certain groups of migrants such as ASs or undocumented migrants, but is also due to barriers that extend beyond the constraints on the legal entitlement to care [15]. Many factors may deter seeking care: a lack of knowledge of the national language, unfamiliarity with the healthcare system, administrative obstacles, and discrimination [16,17], and poverty, too, when user fees are demanded [18]. These factors can lead to unequal access or even to exclusion from health services [10,11,12,13,14,15,16,17,18,19].

Promoting the health of ASs, refugees and migrants has been highlighted as part of the global architecture of universal health coverage [20], and the European regional office of the World Health Organization (WHO) published a strategy and action plan in 2016 [21] in order to address the challenges of the European refugee crisis. The action plan contains recommendations for Member states to strengthen the health system’s capacity to respond to migrants and asylum seekers’ health needs at the arrival phase as well as long term [22,23,24]. The public health of refugees and migrants cannot be separated from the public health of the population and reflects the urgent need for the health sector to more effectively address the impact of migration and displacement on health [25]. Priorities for Member States are the adoption of relevant international standards and policies on refugees’ and migrants’ right to health, both in national law and in practice, and addressing social determinants of health through multisectoral public health policies. 

In Italy, the foreign-born population is about 8.8% of the total population, and migration must be considered a structural phenomenon. In 2019, Italy reported an estimated 354,700 refugees including ASs, accounting for 5.7% of the total number of migrants [26]. According to Italian legislation, after being hosted in hotspots located near borders, ASs willing to apply for refugee status are sent to large reception centers called “hubs”, where they are expected to stay for a maximum of 30 days while their asylum claim is processed [27,28]. Subsequently, they are transferred to the official structures pertaining to the specific Italian system designed for the protection of ASs and refugees (SPRAR system), which offers accommodation and some integration services aimed at guaranteeing protection and facilitating integrated reception at the community level [29]. Given the large number of migrants coming to Italy over the last several years, additional accommodation centers (CAS) have been authorized to accommodate ASs while they wait for a response to their application. [30]. This national architecture responds to and aligns with European policies after 2014 and recent migratory pressures. At the regional level, regional health services (RHS) implement these policies differently, resulting in diverse and unknown models of healthcare.

The Italian Health Service (Servizio Sanitario Nazionale, SSN), founded in 1978, was reformed in the 1990s in a corporate sense, and the constitutional amendment law of 2001 specified how health is a “concurrent” matter between states and regions. Migration, on the contrary, is a matter of state competence, ensuring that the health of migrant populations is a critical element in the organizational relationships between the state and regional levels. The state, therefore, define the general regulatory framework, while the regions establish the operational procedures and laws for implementing policies, in accordance with the national legal framework [31,32]. Each region is divided into Local Health Organizations (LHO), each one with legal and organizational autonomy.

Italian national law guarantees universal healthcare coverage for ASs and refugees, as it does for Italian citizens [33]. Immediately after the regularization of the asylum request, ASs are entitled to be regularly registered in the SSN. Before the submission of their application for refugee status, migrants receive medical assistance managed by local administration. Due to this decentralization, some variations in healthcare offered across the regions are possible [34]. In 2017, the National Health Institute (Istituto Superiore di Sanità—ISS) along with other Italian scientific societies, released the first national guidelines on how to deal with health issues of migrants and ASs in each migratory phase, in order to homogenize the assistance in each region [35,36]. It recommends tackling communicable (CDs) and non-communicable diseases (NCDs) as well as detecting other vulnerable conditions. In addition, it has been suggested to organize more inclusive healthcare services through the appropriate training of personnel and specific care pathways for migrants. Health policies are full-fledged social determinants of health: they can influence aspects of the delivery of health services and the accessibility of health services, which thereby affects health outcomes [37].

In recent years, several studies have sought to analyze policies towards migrants’ access to healthcare with the production of analytical frameworks [23,38]. Accordingly, the management of the arrival phase—as well as that of the following phases—is key for protecting the health of ASs. However, the lack of information about policy implementation and health outcomes has made it difficult to evaluate the experience of ASs on the ground, since it is affected by the presence or absence of a government policy [39,40,41]. Moreover, no evidence has been provided yet in terms of description, continuity, and comparison in reference to international guidelines.

The objective of this paper is to thoroughly examine the policies, practices and care pathway of ASs upon their arrival in the regions, in order to assess the regional and local health system responses in Italy during the European refugee crisis. To the best of the research group’s knowledge, at the time of writing of this paper, no one has explored this field by paying particular attention to the policies and practices. 

Our more specific aims are: (1) to analyze the regional health policies tailored to ASs healthcare at arrival in the various Italian regions, and compare these with national and international recommendations; (2) to evaluate the practices at the LHO level, in order to identify models of care and challenges, and to assess the local implementation of the corresponding policy. 

## 2. Materials and Methods 

### 2.1. Study Design and Setting

This is a multicentric descriptive and comparative qualitative study of migrant healthcare policies and models of care, specifically targeted at ASs in the first period after their arrival in Italy. A content analysis of the policies at the regional level was performed and data regarding local practices were collected within four Italian regions (Emilia-Romagna, Lazio, Toscana and Veneto) since the end of 2017 to present, in coherence with the Italian SSN organization and with the competences of each level, as previously discussed.

### 2.2. Data Collection

A preliminary context analysis of the migratory phenomenon within the investigated regions was performed with regards to the history of migration, quantification of the AS population and a description of each regional reception and accommodation organization. Then, a multiple phase methodology of data collection was used according to the following steps. 

#### 2.2.1. Policy Collection

The subject of the present collection and analysis was regional legislation [42,43]. Only policies enacted at the regional level, since the end of 2017 and focused on migrant and AS healthcare, were included in the analysis.

An organic search on the Internet was performed using institutional sources such as Wikinmp, regional official websites, and the website of the Italian Society of Migration Medicine (Società Italiana di Medicina delle Migrazioni, SIMM) searching for laws, decrees or plans specifically addressing ASs. Public health residents of the research groups in Italian regions included in the study collected the information. The same researchers analyzed and compared data from the policy collection and interviews. Policies at the national level provide the regulatory framework of the research and they were excluded from this analysis [33,44,45,46]. Similarly, regional policies addressing social determinants of health or integration issues were not considered.

#### 2.2.2. Data Collection of Practices 

Data regarding practices were collected through a checklist (Appendix A). It investigated several aspects of medical care, such as first medical examination (ME), immunization, screening programs, as well as other aspects related to care provision and treatment pathways for ASs before their legal entitlement. 

This evaluation followed a fixed workflow. Within the LHOs, directors of health departments and health professionals of migrant healthcare organizations and care provision were approached. At least one healthcare worker for each LHO was approached, for each LHO in each of the four regions. Then, researchers sent the checklist to explain how and why to insert information. Some instructions and technical specifications were sent in attachments to support the compilation. A semi-structured interview was carried out with these privileged observers to deepen our understanding of models of care, identify barriers to access and critical issues, and map the real organization of services with the broadest coverage and to describe pathways, strengths, and weaknesses. The information collected was carried out using the protocols of the single LHO. The process was completed in 2018. A large amount of information (contacts, protocols, challenges) was also obtained thanks to the cooperation with members of regional immigration and health groups (GrIS) of the Italian Society of Migration Medicine (SIMM). GrIS is a local network of health professionals and other members of civil society, with the aim of sharing knowledge, competencies, contacts, relationships, experiences, and practices on migrant healthcare. This organizational model is not a formal institution, but a meeting place, where voluntary participation varies according to the needs expressed by the participants themselves. 

### 2.3. Data Analysis

#### 2.3.1. Policy Content Analysis 

Based on an existing framework specifically designed for migrant health policy analysis [23], the research group developed a dedicated tool for the systematic comparison of policies to better focus on the population of interest. According to international and national recommendations [21,35,36], Mladowsky’s framework was adapted, introducing further categories and items specifically targeting ASs and refugees. In fact, the macro-categories (population groups targeted; data collection and research; quality and accessibility of health services; health issues addressed and implementation) were modified to define new subcategories and to integrate the international and national recommendations. This was done because the present study aims to focus on ASs and refugees upon their arrival in the regions, by comparing policies and their implementation with existing guidelines. The following macro-categories were used for the analysis [23,39]:Data collection assesses action aimed at supporting a migrant-sensitive data collection system such as the computerization of data during all the phases within ordinary data systems. Moreover, it includes other aspects of data collection, such as the aim of the collection and the typology of data collected.Population groups refers to the subtype of migrant population included or specifically outlined in the policies. The analysis was limited to ASs and refugees in the first period after their arrival in Italy, specifying when policies contain indications towards specific categories such as unaccompanied and separated children (UASC), pregnant women, adolescents, the elderly, people with disabilities, people with mental health issues and victims of violence and torture.Health issue addressed considers the health conditions receiving specific attention and any recommendations in the plan and policies analyzed, and it aims to identify the definition of actions specifically directed towards certain high-burden health problems, such as screening, treatment and follow up for CDs, immunization, screening, treatment and follow up for NCDs, screening and multidisciplinary diagnostic–therapeutic–rehabilitation paths for vulnerabilities, maternal and child health (MCH), counselling, health education and health promotion.Part of health system targeted outlines specific actions concerning the organization of healthcare services, such as overcoming barriers in access to care, reinforcing comprehensive primary healthcare and health promotion, improving monitoring and governance and providing training, guidance and support to implement migrant-sensitive interventions.

Table 1 reports a summary of the process from the macro-category to the items investigated, through to the collection of recommendations for each category (a more detailed framework is reported in the Appendix A).

#### 2.3.2. Analysis of Practices

Data collected from interviews were transcribed in a single checklist and then reported in a unique spreadsheet following the checklist framework. Quantitative data were analyzed with statistical descriptive analysis, while qualitative data were analyzed through textual analysis and framework methods [47]. Each researcher collected and analyzed data from his or her regions of work. Data collected on the practices from each region were compared with regional policies. The results were shared and discussed with the other researchers.

## 3. Results

### 3.1. Analysis of Policies

The first survey permitted the collection of regional policies that include ASs healthcare (Appendix A) in four regions of Italy (Emilia-Romagna, Lazio, Toscana, Veneto) [48,49,50,51,52,53,54,55,56,57,58,59,60,61,62,63,64,65,66,67,68,69,70,71,72,73,74,75,76,77,78,79,80,81,82,83,84,85,86,87,88,89,90,91,92,93,94,95,96,97,98,99,100]. Different kinds of policies were found: regional laws and decrees, plans, protocols and guidelines. The entire process of policy analysis is reported for each single macro-area of Table 1 in the Appendix A.

Concerning data collection, in all the regions that were investigated, the collection of data about healthcare at arrival is strictly recommended (Appendix A). However, only one region has planned to implement a continuative and computerized migrant-sensitive collection system and has invested in the portability of data from the first ME. Similarly, qualitative surveys and annual reports on migrants’ health status are lacking in most of the regional policies analyzed. 

Several regional policies are aimed at specific population groups (Appendix A). Access to care for ASs and their right to health are formally recognized by all regions investigated through the validation of the asylum request, which gives the right to be enrolled in the RHS and, as a consequence, to have the same rights as the host community, concerning diagnosis, treatment and preventive services, with special attention paid to maternal and child health. Health protection for elderly immigrants, on the contrary, has been regulated only in two regions, alongside healthcare for migrant with disabilities or with mental health issues, and victims of violence (only in Emilia-Romagna).

Regarding policies targeting specific health issues (Appendix A), policy analysis shows that each region regulated the healthcare of ASs at arrival, identifying practices and protocols that LHOs must follow. First ME, syndromic surveillance, screening for CDs, immunization programs, despite some minimal differences, are expected in all the regions surveyed. However, our analysis revealed the lack of specific policies and protocols for screening and care for NCDs, vulnerabilities and drug abuses, and health promotion.

Policies with specific targets in the health system were collected in each region (Appendix A). All the regions have developed, during the last 20 years, dedicated services for migrant healthcare and linguistic and cultural mediation services targeted at the whole immigrant population, including ASs. Even if governing and monitoring policy contexts are widely present, only two regions have predisposed a specific government focal point for migrant health, with the aim of evaluating regional policy implementation at the LHO level and coordinating the various stakeholders involved in AS healthcare. Concerning training and technical support, only three regions have planned to provide technical guidance to LHOs and healthcare workers (HCWs).

### 3.2. Analysis of Practices

The second survey permitted the collection of data concerning practice at the LHO level. Thirty-four checklists were collected, reaching 65% of the total amount of LHOs present in the regions included in the study (Table 2). Covering all of the LHOs present in every region was not possible only in two regions (Lazio and Veneto) because of the logistical difficulties faced by the researchers. Different kinds of HCWs were involved in the survey as LHOs’ referees for AS healthcare, such as health managers (HMs), nurses, medical doctors (MDs).

Each LHO provides the first ME, tuberculosis (TB), sexually transmitted disease (STD), scabies and pediculosis screenings and treatments 2 to 3 days and 6 months after arrival (Table 3). Screenings for other CDs (Human Immunodeficiency Virus (HIV); hepatitis B virus (HBV); hepatitis C virus (HCV); latent tuberculosis infection (LTBI)) and for NCDs appear to be provided with more variably between LHOs, occurring during the first ME at the arrival or during the general practitioners’ intake after entitlements to the RHS have been confirmed, depending on the model adopted by the LHO or by the timeframe for entitlement to RHS. Data from the first ME, screenings, and other investigations are collected in all the LHOs, but the data collected are not continuative with the collection ordinary systems, and only one regional system has implemented a regional computer system. 

Access to health services for diagnosis, treatment and follow up for CDs and NCDs are formally guaranteed in all the LHOs without cost for the first six months after arrival (12 months in Tuscany) and the entitlement to RHS is possible after the formalization of an asylum request. Moreover, maternal and child healthcare are provided in all LHOs, and some LHOs have activated multidisciplinary diagnostic–therapeutic–rehabilitation paths for people with vulnerabilities.

From our analysis of the local care pathway, three main models emerged, which are widely distributed in the regions investigated. 

The first model consists of the presence of dedicated services for migrants including ASs, provided by LHOs. They offer healthcare to ASs from their arrival until their enrolment in the RHS. Another model bases the central role of the general practitioners in ASs’ care. The guarantee of access lasts from the moment of arrival until enrolment in the RHS, as in the previous model. The third model refers to the involvement of non-governmental organizations (NGOs) by LHOs in providing healthcare to vulnerable populations. As in the previous models, healthcare is provided from arrival until enrolment in the RHS. Other models of AS healthcare are more fragmented and based on the presence of different actors, such as physician involvement in the reception center, or public health professionals in the LHOs. According to the interviews, these models only focused on syndromic surveillance and they are not related to proper early intake care, even if access to services in case of need is guaranteed. 

Our analysis of practices at the LHO level reveals several challenges. First, in most of the LHOs investigated, HCWs reported the presence of different kinds of barriers in access to care for ASs, dependent on linguistic–cultural factors and on legal status (Table 4). In particular, the entitlement to the RHS seemed to be impeded to the expiration of valid permission to stay or the delay of asylum requests. 

Other challenging areas, reported by HCWs, are the screening and treatment for NCDs and dental health. Similarly, healthcare paths for people with vulnerabilities have been described as fragmented and inefficient, along with the continuity of care from arrival through to the next phases. The computerization of data, data collection and transmission were not well implemented, both where we expected they would be and where we did not. In all regions, health assessments of ASs and the monitoring of policy implementation, where expected, are not regularly conducted. 

Difficulties in coordination between health professionals of LHOs and other stakeholders (law enforcement, NGOs, social services), as well as the lack of integration between different departments and HCWs of LHOs (primary care departments (PCDs); public health departments (PHD), general practitioners (GPs), hospitals) have been widely reported. Other issues that have been reported concern the training of health professionals, which seems to be insufficient and not regularly conducted, especially for general practitioners and health professionals involved in assistance at arrival. However, in several LHOs, training for HCWs has been provided by specific projects funded by the financial programs of the EU for migrants and ASs (European Asylum, Migration and Integration Fund (AMIF)), and managed by RHS and LHOs.

Finally, health promotion was described as one of the weakest areas of the healthcare of ASs provided by LHOs, and most of the activities related to this field have been managed by NGOs, civil society organizations (CSO) and by the spontaneous networking of health professionals.

## 4. Discussion

This comparative content analysis of regional policies and practices towards ASs allowed for a broader examination of the gaps between regional policies and international and national recommendations, as well as the implementation of policies at the local level and the main model of healthcare adopted by the LHOs. Moreover, it allowed for the assessment of regional and local health system responses and the identification of challenges and best practices. It produced both an overview of data and an analysis within the regions and, finally, a comparison between them. 

Concerning regional policy, the results showed that specific policies targeted towards AS healthcare in response to the recent migratory phenomena have been regulated in all the settings investigated. Additionally, results highlighted that policymakers gave more attention to the first phase of healthcare at arrival, syndromic surveillance, and infectious disease screening and control, than to the other phases and aspects [101]. From the data collected, this period appears to be rule based and particularly homogenous across the regions. However, some areas seem to be neglected by regional policy frameworks, such as the screening of NCDs and health protection for people with vulnerabilities. Consequently, actions and solutions aimed at responding to the first phase lead to fragmentation [102].

Our policy analysis demonstrated the presence of many items of international and national recommendations in regional policies, mostly concerning the right to health and access to care, CD prevention and control, health protection of mothers and children, cultural mediation services, and migrant-friendly services. Most of these policies are in line with national and international recommendations, but they seem to be inefficiently implemented. Specifically, results from different LHOs showed that data collection and computer systems, cultural mediation services, training and technical support, as well as monitoring and health assessment, are the weakest areas. 

Evidence from migrants’ health status upon arrival showed a great burden towards NCDs, complex health needs and vulnerabilities [103,104,105,106,107,108,109,110]. This is a consequence of displacement and the migratory process and social determinants of health in the host country [111]. While generalizations are not possible, a few key points regarding intake into a continuous and integrated healthcare system responsible for the person from arrival emerged. The inclusion in a strong primary care system appears to reduce the healthcare burden [112,113] and reduce costs [114]. Moreover, it is worth noting that greater attention is paid to ensuring access and the right to health after the regularization of asylum [39,40,41,42,43,44,45,46,47,48,49,50,51,52,53,54,55,56,57,58,59,60,61,62,63,64,65,66,67,68,69,70,71,72,73,74,75,76,77,78,79,80,81,82,83,84,85,86,87,88,89,90,91,92,93,94,95,96,97,98,99,100,101,102,103,104,105,106,107,108,109,110,111,112,113,114,115], when ASs gain the same health rights as the general host population, within the universal coverage of the SSN. However, this is not within the scope of the present study.

The analysis of practices showed the full implementation of some aspects of regional policies, particularly the first ME, screening for CDs and immunization programs, as well as cooperation and collaboration with the local office of the Ministry of the Interior (Prefettura) by LHOs and managing bodies of reception centers. In general, concerning these health issues, the practices seem to be similar across LHOs of different regions. 

Screenings for NCDs, health promotion activities and screening–diagnostic–rehabilitation healthcare paths for people with vulnerabilities have been developed only by some LHOs, as a part of a local policy or as a consequence of the availability of resources. 

Many items that were found to be missing through content analysis do not always lack the matching of these indications and guidelines both at national and international levels. The adoption and implementation of directives and policies leads to disparities and delays, as described, and policies may not be adopted even when implementation practices come before them. 

Our analysis of practice also revealed several challenges at the local level. Linguistic–cultural, administrative, and legal barriers to accessing care [102] are present across the regions, reflecting a lack of cultural competence in the health service [116,117,118]. Access to care depends on legal status [14,15,16,17,18,19,20,21,22,23,24,25,26,27,28,29,30,31,32,33,34,35,36,37,38,39,40,41,42,43,44,45,46,47,48,49,50,51,52,53,54,55,56,57,58,59,60,61,62,63,64,65,66,67,68,69,70,71,72,73,74,75,76,77,78,79,80,81,82,83,84,85,86,87,88,89,90,91,92,93,94,95,96,97,98,99,100,101,102] as well as proper response to specific needs, such as vulnerabilities [102]. These observations reveal the lack or the ineffectiveness of interventions aimed at overcoming these barriers and at building migrant-friendly services, focused on the migrants’ health needs and oriented to guaranteeing equitable, acceptable and adequate care [119,120,121] both at arrival and long term.

Other challenges reported in the interviews concern the lack of integration and cooperation between different departments of LHOs, and difficulties of continuity of care [102]. These aspects are already known as some of the main weaknesses of the current organization of the LHOs and are associated with low quality of care [115]. In migrants’ and ASs’ healthcare, the presence of barriers to access, in addition to difficulties in continuity and integration of care, can produce delays in diagnosis and treatment, inadequate care or overtreatment and disparities in health outcomes [119,120,121,122]. This requires that health service organizations play a central role in guaranteeing ASs’ right to health and their health protection, especially at the primary healthcare level, where a migrant-friendly approach seems to be more effective in reducing healthcare burdens [112,113]. 

Since primary healthcare seems to play a central role in migrant healthcare, the analysis of the models of care that emerged from the analysis of local paths can offer some support to the debate on how to strengthen health services to address migrant and AS health needs. 

The presence of dedicated services on migrant healthcare seems to reduce the risk of ASs being excluded from healthcare or of inappropriately using emergency services [123]. This is a consequence of a lack of knowledge about the healthcare system, mostly in the first period after arrival, and a lack of health coverage due to a delay in the processing of an asylum request, especially during periods of large influx [112,113]. Moreover, these services seem to have specific migrant health competences, while other RHS do not. In spite of this, the presence of dedicated services could reflect the risk of developing separate care pathways both for migrants and host communities [123] that could seem discriminatory. In addition, the presence of dedicated services could impede ASs’ empowerment in accessing and taking care of their health and reduces the responsibility of other services, especially of general practitioners.

The second model is based on the early intake care of ASs by general practitioners, which are the entry point of primary healthcare. As in the previous model, this one provides health protection from arrival, but it reduces the risk of creating different pathways of care for migrants and host communities [112,113]. However, according to the interviews, GPs within LHOs either lack specific training in migrant health or benefit from it disproportionately, and mediation services outside LHO facilities are not widely available.

Moreover, early intake care of ASs by GPs can result in delays or inappropriate responses to the health needs of ASs, especially in the case of complex needs and when GPs are not particularly skilled in migrant health [124].

Another model emerged regarding the presence of NGOs. The results showed that HCWs of NGOs are usually particularly skilled in migrant health and complex health needs. However, the risk remains in dividing the care pathway and of outsourcing some services from the SSN. In fact, many of these NGOs are not fully integrated with the network of health services and have no access to ordinary data systems. Furthermore, as stated, LHOs and RHS are not able to assess ASs’ and migrants’ health needs attended to by these NGOs. Moreover, the position of NGOs is complex and cannot be generalized. In fact, NGOs must navigate the delicate balance between providing assistance and healthcare and advocating and conditioning public system decisions [125]. 

This variability in models of care adopted by LHOs can have more than one explanation: the regional autonomy towards healthcare and the decentralization to a subnational level of health policies and health management, in coherence with the Italian SSN organization and with the competences of each level [32,33]; the characteristics of the settings, such as the availability of resources or the previous policy [123]; geographical (urban or rural area), cultural or political aspects. In this study, the previous migratory pressure on RHS and the organizational model of the reception systems seemed to be more influential, even if more data and details are needed to better understand the possible correlation between these factors. 

Since different contexts require different policies and models of care [39,112,113], extensive variability and heterogeneity are not wholly negative, but have positive aspects as well. On one hand, this variability could risk of hiding the gap between the policies and their implementation and the guidelines at the international and national levels. On the other hand, it could act as an engine of change with attention to the process of networking. In this regard, governance and monitoring play a key role and represent a serious lack in the health system. 

The critical issues summarized above underline the need to reinforce the national and regional levels of governance towards ASs’ and migrants’ health [21,22,23,24,25]. A migrant-friendly approach and a stronger primary healthcare system could be a part of the solution, although it means integrating governance systems capable of challenging all the actors involved and building a community-based approach. 

An essential component of building an inclusive system is to identify and constantly monitor the compliance with international recommendations and national policies and guidelines, while assessing the health needs of ASs and migrants. Accordingly, critical areas emerged from the analysis that demonstrated the importance of monitoring policy implementation. Therefore, the design and commissioning of a permanent observatory towards policies represents a useful monitoring and governance instrument [25]. 

Concerning the latter, it is worth noting that local resources and practices, in many cases, bridge the gap, offering sustainable and inclusive practices. Moreover, many European-financed programs, as well as national and regional projects specifically devoted to ASs, fill the gaps in policies. From the interviews, it emerged that informal practices are often successful. The main factors of this process are the multiplicity of actors of the community and the local social structures, such as NGOs or volunteering, which add value to local resources and practices, act to reinforce networks and contribute to cohesion and social integration [115]. The adoption and implementation of directives and policies lead to disparities and delays, as explained, and policies are occasionally not adopted, even when implementation practices come before them. 

The present research presents many strengths and limitations. The main limitations are: The partial coverage of the country. In fact, the study included only four Italian regions, located in the north and center of the country.The incomplete data concerning the sources of information. Documents collected in the policies category did not consider/include projects and programs financed using European funds (such as AMIF projects).The large amount of data collected. This made the analysis complex and some simplifications of the details were needed.The lack of similar policy analysis studies as frameworks and tool comparisons, meaning that we needed to mix and match the Mladovsky tool [23] with national and international guidelines and with the difficulty of policy implementations.

The main strengths are: The relevance of the work, because of the scarcity of such research design methodologies, and few data sources. Therefore, the collection and production of data analysis on the topic appear to be relevant, as does the description of the gap referring to the international and national guidelines. These factors seem not to be present in similar studies associating policy levels to policy implementation.The originality of both the study design (descriptive and comparative study, using a mixed methodology, both content analysis of policies and survey and semi-structured interviews with health workers to evaluate implementation), and the data collection (collaboration with local immigration groups, e.g., GrIS; proximity and closeness to the territory of the research group members; identification and engagement of key stakeholders; methodology of action–research–intervention).The definition of a multicentric and multi-located research project, involving a regional and sub-regional detail level, unlike other similar studies [39,40,115].

## 5. Conclusions

This study describes some aspects of the health system response to the recent migration phenomena in four Italian regions. Regional policies addressing migrants’ health mainly focus on the prevention and control of infectious diseases upon their arrival, while less attention is given to NCDs and to taking charge of vulnerable groups. Access to care for migrants and asylum seekers seems to still be hampered by some barriers and factors, other than those recommended by regional policies and national/international guidelines. Local realities on the ground have filled these gaps thanks to some best practices based on the strengthening of the primary healthcare services and the involvement of local governance and other stakeholders. 

Since it appears that there is no one-fits-all healthcare model, each region has to adapt the healthcare policy to their specific local context. This requires interdisciplinary and intersectoral actions, engaging not only the healthcare sector, but also other local stakeholders, such as NGOs or volunteering organizations, while also involving other local resources and practices, reinforcing networks and contributing to cohesion and social integration. However, local policy implementation may lead to different types of healthcare assistance and then threaten equity and quality in healthcare for migrants. Therefore, the policies that we believe to be the most desirable directly target the empowerment of primary healthcare services for migrant assistance.

## Figures and Tables

**Table 1 ijerph-17-05458-t001:** Framework for Policy Content Analysis.

Area	Summary of Recommendations	Items/ Target Investigated
Data collection	Identify immediate needs during episodes of mass international migration; Use informative system to collect data during all the phase of the reception. Promote the inclusion of migrant variables in existing data collection systems; Use of a defined checklist/protocol for medical examination; Inclusion of AS data in the ordinary data system; Computerization of data.	Continuative and computer migrant-sensitive collection system
	Collection of and access to information on the health status, modifiable risk behaviors and access to healthcare; continuous health needs assessment. Disaggregation and comparability of data is required; Enhance epidemiological surveillance capacities to include migrant-sensitive data. Use innovative approaches, including surveys and qualitative methods.	Typology of data
	Promote the portability of health data in accordance with national law.	Portability/Transmissibility
	Produce progress reports on the health status of refugees, asylum seekers and migrants.	Report/scope of collection
Population groups	Improving the health of the most vulnerable, including unaccompanied children, pregnant women, adolescents, the elderly, people with disabilities and victims of torture. Issues relating to sexual and reproductive health, family planning, gender-based violence and rape management, forced marriage and adolescent pregnancy, and mental health and care should be prioritized.	Unaccompanied children (UNCH) Pregnant women Adolescents Elderly People with disabilities People with mental issue Victims of violence (any) and torture
Health issue addressed	Screening during first ME: TB, malaria, STDs, parasitosis; Screening: HIV, HBV, HCV, LTBI; Involvement ASs in infectious disease prevention and control;	Screening, treatment and follow up for CDs
	Immunization programs for children (0-14) and adults (polio, diphtheria, tetanus, pertussis, measles, mumps, rubella, chicken poxs, HBV)	Immunization
	Screening, early access to essential primary care, accessing treatment, care and support; Screening for visual and auditory acuity, dental health, diabetes, hypertension, anemia, cervical cancer; blood tests: blood count, urine test;	Screening, treatment and follow up for NCDs
	Screening for psychosocial disorders, drugs and alcohol abuse, nutrition disorders; Screening for violence and torture, specific and multidisciplinary diagnostic–therapeutic–rehabilitation path;	Screening and multidisciplinary diagnostic–therapeutic–rehabilitation path for vulnerabilities
	Screening for pregnancy, access to screening programs that are in place for the host population, screening during pregnancy for neonatal diseases, access to maternal and neonatal healthcare	Maternal and Child health
	Counselling, health education and health promotion	Counselling, health education and health promotion
Part of health system targeted	Culturally sensitive health services, access to interpreters, provision of cultural mediators; Overcome administrative hurdles; Support for patient fees; Information about health entitlements and support in navigating through the system;	Overcome barriers in access to care
	Primary care, preventive care, health promotion services, prevention, detection, treatment and monitoring of NCDs, CDs, vulnerabilities, MeCH;	Comprehensive primary healthcare and health promotion
	Health assessment; Reporting of implementation, accountability and data collection; Government focal points, cooperation and coordination with other stakeholders; Community information and engagement;	Monitoring and governance
	Training and continuous update with health equity and human rights-based approaches, and specific focus (es. victim of torture); Skilled health professional on migrant health/continuous professional training; Guidance, training and support tools to implement migrant sensitive interventions on CDs, NCDs, vulnerabilities;	Continuous training, guidance and support to implement migrant sensitive interventions

Note: Asylum seeker (AS); communicable diseases (CDs); Human Immunodeficiency Virus (HIV); hepatitis B virus (HBV); hepatitis C virus (HCV); latent tuberculosis infection (LTBI); medical examinations (ME); maternal and child health (MCH); non-communicable diseases (NCDs); sexually transmitted diseases (STDs); tuberculosis (TB); unaccompanied and separated children (UASC).

**Table 2 ijerph-17-05458-t002:** Summary of Checklists.

Area	Emilia-Romagna	Lazio	Toscana	Veneto
LHOs covered	9/9	5/10	3/3	3/9
Checklist	14	5	12	3
HCWs	HMs, nurses, MDs;	HMs, MDs;	HMs, MDs;	HMs, MDs;

Note: Health Managers (HMs); Local Health Organization (LHO); Medical Doctors (MDs).

**Table 3 ijerph-17-05458-t003:** Summary of Checklists: Practices at the LHO level.

Area	Emilia-Romagna	Lazio	Toscana	Veneto
Reception system	Hub-Spokes	Mixed	Widespread	Widespread
First ME when who where why what	2-3d–1m; DS of LHO, NGOs, GP; LHO and NGO facilities, reception center; Syndromic surveillance and active research of health issue;	2-3d–1m; DS of LHO, NGOs; GP; LHO and NGO facilities; Syndromic surveillance and active research of health issue;	2-3d–1m; MD of LHO; NGOs; LHO and NGO facilities, reception center; Syndromic surveillance and active research of health issue;	2-3d–15d; PHD, DS of LHO; PHD, DS of LHO; Syndromic surveillance, and active research of health issue;
Screening when who where why what	2-3 d–6 m; PHD, IDU, DS of LHO; PHD, IDU, DS of LHO; Individual and community health protection; TB, TSD, parasitosis, pediculosis, scabies, LTBI, HIV, HBV, HCV, vulnerabilities;	2-3 d–6 m; PHD, IDU, DS of LHO; PHD, IDU, DS of LHO; Individual and community health protection; TB, parasitosis, vulnerability;	2-3 d–6 m; PHD, IDU, PCD of LHO PHD, IDU, PCD; Individual and community health protection; Not everywhere: TB, Syphilis, HCV, HBV, HIV, vulnerabilities;	2-3d–1m; PHD, DS of LHO; PHD, DS of LHO; Individual and community health protection; TB, LTBI, scabies and Polio;
IP when who where why what	1m–6m; PHD; PHD; Individual and community health protection; National programs;	1m–6m; PHD; PHD; Individual and community health protection; National programs;	1m–6m; PHD; PHD; Individual and community health protection; National programs;	1m–6m; PHD, DS of LHO; PHD, DS of LHO; Individual and community health protection; National programs;
Data collection	Computerized regional system and papers;	Paper checklist	Paper checklist	Paper checklist
Access to care	Free of charge for first 6 months;	Free of charge for first 6 months;	Free of charge for first 12 months;	Free of charge for first 6 months;
After first ME when who where why what	Until entitlement to RHS; NGOs, DS of LHO, GP; Lack of orientation, complement of CDs screening; Access at need, take in care, CDs screening and treatment;	Until entitlement to RHS; NGOs, DS of LHO, GP; Lack of orientation, complement of CDs screening; Access at need, take in care, CDs screening and treatment;	Until entitlement to RHS; NGOs, DS of LHO, GP; Screening completion, answer to acute health needs and take care of chronicity and frailty;	Until entitlement to RHS; PHD, DS of LHO, NGOs; Lack of orientation, delay on entitlements to RHS; Access at need, take in care, CDs screening and treatment;

Note: Communicable diseases (CDs); dedicated services for migrant health (DS); general practitioner (GP); Human Immunodeficiency Virus (HIV); hepatitis B virus (HBV); hepatitis C virus (HCV); infectious diseases units of LHOs (IDU); Local Health Organization (LHO); latent tuberculosis infection (LTBI); medical examination (ME); non-communicable diseases (NCDs); non-governmental organization (NGO); primary care department (PCD); public health department (PHD); regional health service (RHS); sexually transmitted diseases (STDs); tuberculosis (TB); unaccompanied and separated children (UASC).

**Table 4 ijerph-17-05458-t004:** Summary of checklists: challenges.

Area	Emilia-Romagna	Lazio	Toscana	Veneto
	Access to dental healthcare; Data transmission; Cultural barriers in access to care; Delay on entitlements to RHS; Lack of multidisciplinary diagnostic–therapeutic–rehabilitation path for vulnerabilities;	Data collection and transmission; Delay on entitlements to RHS; Integration DS of LHO and GP; lack of cultural mediators; Lack of multidisciplinary diagnostic–therapeutic–rehabilitation path for vulnerabilities;	Lack of a computerization of health data; Heterogeneity in models of care and practices; Lack of the continuity and inclusivity of care after the first ME with the health system;	Lack of a computerization of health data; Heterogeneity in models of care and practices; Lack of communications between stakeholder involved in take in care; Delay on entitlements to RHS; Lack of orientation; Lack of specific services;

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
