# Peer review of "Health System Response during the European Refugee Crisis: Policy and Practice Analysis in Four Italian Regions"

_ijerph, 2020, doi:10.3390/ijerph17155458_

Round 1

Reviewer 1 Report

This paper addresses an important topic in public health, namely health access of asylum seekers (ASs) as a particularly vulnerable group of migrants. The paper makes an important contribution to the literature and thus, I strongly recommend publication with the following minor revisions.

The literature review could be expanded to include more recent studies on refugees' and ASs' health access, with regards to the latest inflow of forced migrants following the great summer of migration on 2015. I would also like to see the results of the paper embedded into the more specific debate on health access and health rights of AS, rather than migrants in general. The EU guideline on reception of AS should be elaborated on to provide some context to readers not familiar with the EU policy framework.

Provide reasoning why exactly these four Italian regions were chosen and not others (might be for purely logistical/pragmatic reasons, but do state clearly in the text).

I missed a reference to research ethics and in how far they were adhered to when contacting LHOs. Was there a research commission involved in confirming the study design and work plan?

The conclusion should be extended to include a more throrough, specific discussion of results. What exactly can we infer from the findings? Also, how can the results be upscaled? How can they be applied to other geographical contexts, i.e. what can other countries learn from these insights? I'd also like to see some concrete policy recommendations following from the conclusion.

Reviewer 2 Report

reduce acronyms 

tables ...confusing contents/layouts

248, for example, needs clarity/rewrite

same for 274, 295, 298

328-what interviews? Need clarity and details

298 - what is in parenthesis? How is it relevant?

295 - some point? Provide a date

405 - indication means?

411- limits vs. limitations?

Reviewer 3 Report

The article addresses an important issue, and it is clear that the researchers have done an extensive job of data collection. However, I think more analysis and/or a major revision of the manuscript is required before it can be of benefit to an international audience.

A first problem is that the authors present their aims in different ways in different parts of the manuscript, so it is difficult to assess if the methods, results, discussion and conclusion are in line with the objectives. For example, in p. 3 the objective is stated as “... to thoroughly examine policies, the connection between policies and their implementation and the care pathway at arrival in the regions”, and also as “to define a useful context-specific evidence and model of care by evaluating regional and local responses to the recent migratory phenomena”. These are different things, and I recommend the authors homogenize their statement of the objectives.

That said, if I understood correctly there are two main objectives of this article: 1) to describe the regional policies regarding migrant health in four regions of Italy; 2) to analyze if the practices at health services in those regions conform to those policies (and to international recommendations). However, the results are not focused on those aims. Instead, a description of dimensions of practices (Table 2) is followed by a description of policies and a brief comment of the gaps between those policies and international recommendations, and a detailed description of practices which extends into the discussion. The relationship (or gap) between regional policies and regional practice is not extensively presented nor summarized.

Moreover, what I fell is lacking is an analysis of what can be learned from the description of policies and practices that is of general interest for the study of health systems or access to health care for migrants. Some interesting aspects of this are hinted at, but not developed: what is the relevance of the regional (secondary legislation) in this regard (why did the authors focus on regional, as opposed to national, legislation)? What is the role of governance? Are there reasons that could explain the differences between regions?

The identification of three models of migrant healthcare seems an interesting and useful finding, but it’s not discussed enough. How were these models extracted from the checklists? Are these three models the consequences of three different types of policies? Were the models designed as part of the policies, or how they came to be?

In order to understand the relationship between policy and practice, more context would be useful: are LHOs part of RHSs? Is the reviewed legislation mandatory for both? Are NGOs included in legislation? When the authors mention that “Due to this decentralization, some variations in healthcare offered across the regions are possible”, what kinds of differences do they mean (e.g., are some services covered in a region but not in another, or is it more a matter of service quality)?

Some important information is missing from the methods:

  • Through which search engine was the Internet search performed?
  • I replicated the search “migrant* OR asylum seeker* OR refugee* AND health AND law OR decree OR plan” in Google and over six million results came up, with no links to secondary legislation in regions in Italy in the first page. Did the authors use the regions’ names as part of the search criteria?
  • What were the inclusion criteria for policy documents? All secondary legislation, or only secondary legislation issued by the ministry of health? A working definition of “secondary legislation” might also be useful
  • How was the sample for the second component (study of practices) selected? If there is more than one LHO per region, how were participant LHOs chosen? Is there only one director of health department per LHO? How were other health professionals selected?
  • If healthcare workers were “involved in the collection data process”, is there a potential for bias here, since the same people whose work was being evaluated collected data?
  • “A large amount of information was also obtained thanks to the cooperation with members of regional immigration and health groups (GrIS) of the Italian Society of Migration Medicine (SIMM).” How many people, how were they chosen?
  • The authors say that “the entire process is reported in Supplementary materials, Table S2)”. However, table S2 only presents the items that were assessed to reflect Mladovsky’s dimensions. How was the information obtained and processed? How was the data obtained as described in Table S2 combined with data from the checklist in Table S1?

Finally, I suggest the authors consult with a translator or native English speaker, as some phrases are confusing.

Round 2

Reviewer 1 Report

Thank you for your revisions and additions to the paper, which improved quality and readability of the manuscript and helped to clarify some several key aspects. I hence recommend to accept the paper for publication.

Reviewer 2 Report

Please provide clarity on sample size and whether this is internet research only.

Reviewer 3 Report

The authors made some minor changes and additions in response to my comments, but they didn’t conduct the major revision of the manuscript that I feel is needed. My first concern is still that, while they go over a lot of very interesting and relevant information and insights, the way they present them doesn´t do justice to their work. I think the manuscript still requires trimming in the results and discussion, cutting of repetitive descriptions and lists of items, and focusing in clarifying the main points the authors feel can be extracted from their results. My second concern is that there’s still a lack of clarity in the methods. Specific points are:

  • The authors’ description of their objectives improved, but I still find their justification confusing. When they say “these policies are linked to the possibility of each region to manage its healthcare system and reception system”, what do they mean? In what sense would the policies affect the regions’ possibility of managing their healthcare and receptions systems? This needs to be spelled out.
  • In my previous comments, I made some suggestions about issues that the authors might want to develop, in order to go from the descriptive to the analytical. In response, the authors added some lines about governance (lines 398-408) and differences between regions (lines 400-406), which I think could be the basis of a more interesting discussion (replacing repetitions of the aims and descriptive results in this section).
  • In their response, the authors say they added detail on the relationship between LHOs and RHSs in line 93, but I couldn’t locate that addition
  • There is still no description of the online search criteria. The authors deleted the list of keywords employed in the online search, that appeared in the previous version, why?
  • The following comments were partially answered in the authors’ response letter, but they need to be included in the methods section of the manuscript as well:
    • What were the inclusion criteria for policy documents? All secondary legislation, or only secondary legislation issued by the ministry of health? A working definition of “secondary legislation” might also be useful
    • How was the sample for the second component (study of practices) selected? If there is more than one LHO per region, how were participant LHOs chosen? Is there only one director of health department per LHO? How were other health professionals selected?
    • If healthcare workers were “involved in the collection data process”, is there a potential for bias here, since the same people whose work was being evaluated collected data?
  • In response to my comment about information gathered from GrIS and SIMM (“How many people, how were they chosen?”), the authors only added a description of what the GrlS is. As in the previous point, what I would like to see is a thorough description of selection methods (for both documents and informants), allowing the reader to assess the possibility of bias in the results.
  • The authors provide a brief response to my previous comment “The authors say that ‘the entire process is reported in Supplementary materials, Table S2’. However, table S2 only presents the items that were assessed to reflect Mladovsky’s dimensions. How was the information obtained and processed? How was the data obtained as described in Table S2 combined with data from the checklist in Table S1?” in their cover letter, but haven’t addressed my observation in the revised manuscript.
  • About my suggestion of reviewing the English: I don’t think it’s my place to review the manuscript in this regard (and obviously I’m not a native English speaker myself), but here are some examples of the types of phrases I found confusing:
    • “Regional policy analysis has shown heterogeneity concerning the alignment with international and national recommendations” “HAS SHOWN” WOULD SEEM TO REFER TO PREVIOUS LITERATURE OR RESEARCH, BUT IN CONTEXT IT SEEMS THE AUTHORS ARE REFERING TO THEIR OWN RESULTS.
    • “Many items that lack policy content analysis do not always lack the matching of these indications and guidelines both at national and international levels” THE ITEMS LACK POLICY CONTENT ANALYSIS, OR THE ITEMS WERE FOUND TO BE MISSING THROUGH CONTENT ANALYSIS?
    • “This comparative content analysis of regional policies and their implementation towards ASs” IS IT “IMPLEMENTATION TOWARDS ASs”, OR “REGIONAL POLICIES TOWARDS ASs AND THEIR IMPLEMENTATION”?
    • “Moreover, it is worth noting that great attention to ensuring access and the right to health happens after the regularization of asylum [39, 115], when ASs gain the same health rights of the general host population, within universal coverage of the NHS” GREAT ATTENTION OR GREATER ATTENTION?